# Impact of Liver Metastases and Number of Metastatic Sites on Immune-Checkpoint Inhibitors Efficacy in Patients with Different Solid Tumors: A Retrospective Study

**DOI:** 10.3390/biomedicines11010083

**Published:** 2022-12-29

**Authors:** Madeleine Maugeais, Julien Péron, Stéphane Dalle, Amélie Boespflug, Michaël Duruissaux, Pauline Corbaux, Thibault Reverdy, Gulsum Sahin, Aurélie Rabier, Jonathan Lopez, Nathalie Freymond, Denis Maillet

**Affiliations:** 1Oncology Department, Centre Hospitalier Lyon Sud, Institut de Cancérologie des Hospices Civils de Lyon (IC-HCL), 69495 Pierre-Bénite, France; 2Université Claude Bernard Lyon 1, 69100 Villeurbanne, France; 3ImmuCare (Immunology Cancer Research), Institut de Cancérologie des Hospices Civils de Lyon (IC-HCL), 69229 Lyon, France; 4CNRS, UMR 5558, Laboratoire de Biométrie et Biologie Evolutive, Equipe Biostatistique-Santé, 69100 Villeurbanne, France; 5Dermatology Department, Centre Hospitalier Lyon Sud, Institut de Cancérologie des Hospices Civils de Lyon (IC-HCL), 69495 Pierre-Bénite, France; 6Department of Respiratory Medicine, Groupement Hospitalier Est, Hôpital Louis-Pradel, URCOT, Institut de Cancérologie des Hospices Civils de Lyon (IC-HCL), 69500 Bron, France; 7Centre de Recherche en Cancérologie, 69008 Lyon, France; 8Department of Biochemistry and Molecular Biology, Centre Hospitalier Lyon Sud, Institut de Cancérologie des Hospices Civils de Lyon (IC-HCL), 69495 Pierre-Bénite, France; 9Cancer Research Center of Lyon, INSERMU1052, 69007 Lyon, France; 10Department of Thoracic Oncology, Hôpital Lyon Sud, URCOT, Institut de Cancérologie des Hospices Civils de Lyon (IC-HCL), 69310 Pierre-Bénite, France; 11Université de Médecine Jacques Lisfranc, 42270 Saint-Etienne, France

**Keywords:** immune checkpoint inhibitors, PD1 inhibitors, PDL1 inhibitors, metastatic sites, liver metastases, prognostic biomarkers

## Abstract

Background: ICIs have dramatically improved patient outcomes in different malignancies. However, the impact of liver metastases (LM) and number of metastatic sites (MS) remains unclear in patients treated with single-agent anti-PD(L)1. Methods: We aimed to assess the prognostic impact of LM and MS number on progression-free survival (PFS) and overall survival (OS) in a large single-arm retrospective multicentric cohort (IMMUCARE) of patients treated with anti-PD(L)-1 for different solid tumors. Results: A total of 759 patients were enrolled from January 2012 to October 2018. The primary tumor types were non-small cell lung cancer (71%), melanoma (19%), or urologic cancer (10%). At the time of ICI initiation, 167 patients (22%) had LM and 370 patients (49%) had more than MS. LM was associated with a shorter median PFS of 1.9 months (95% CI: 1.8–2.5) vs. 4.0 months (95% CI: 3.6–5.4) in patients without LM (*p* < 0.001). The median OS of patients with LM was of 5.2 months (95% CI: 4.0–7.7) compared with 12.8 months (95% CI: 11.2–15.1) (*p* < 0.001). Interestingly, LM were not associated with shorter PFS, or OS compared to other MS types (brain, bone, or lung) in patients with only one MS. Patients with multiple MS also had poor clinical outcomes compared to patients with only one MS. The presence of LM and MS number were independent prognostic factors on overall survival. Conclusion: The presence of LM or multiple MS were associated with poorer survival outcomes in patients treated with anti-PD(L)-1.

## 1. Introduction

PD-1 is a key regulator of the threshold of immune response and peripheral immune tolerance. It is expressed on activated T-cells, B cells, macrophages, regulatory T-cells (Treg) and natural killer (NK) cells. The binding of PD-1 to its ligands (PDL1 or PDL2), which are frequently expressed on tumor cells, results in the suppression of proliferation and immune response of T cells. Consequently, the activation of PD-1/PD-L1 signaling serves as a principal mechanism by which tumors can evade antigen-specific T-cell immunologic responses [1].

In the last decade, the advent of immune checkpoint inhibitors (ICIs), such as anti-PD1 or anti-PDL1, has dramatically improved patient outcomes with various advanced cancers [1,2]. Anti-PD(L)1s are either used alone or in combination with anti-cytotoxic T-lymphocyte antigen-4 (anti-CTLA4), targeted therapy or chemotherapy [3,4]. Although the combination of anti-PD(L)1 and anti-CTLA4 are frequently associated with durable clinical benefits, only a small subset of patients treated with single-agent anti-PD(L)1 inhibitors experiences durable responses.

It is therefore critical to identify predictive factors of response. Many potential biomarkers for responsiveness or resistance to ICIs have been explored, such as tumor mutational burden (TMB) and tumor-infiltrating lymphocytes (TILs) [5], but only PDL1 expression is routinely used in first-line non-small cell lung cancer (NSCLC) and in cisplatin-ineligible urothelial carcinoma [6,7]. As a consequence, some clinical features, such as ECOG performance status, are used in routine practice to predict responses to ICIs. In particular, it was previously described that the presence of liver metastases (LM) or the presence of multiple metastatic sites (MS) was associated with a low response rate and poor clinical outcomes in patients treated with chemotherapy or targeted therapy [8,9,10,11]. However, only a few retrospective studies investigated the impact of MS number and type in cohorts of patients treated with PD(L)1 inhibitors [12,13]. Data regarding the impact of MS number and LM on clinical outcomes could be important for decision-making of clinicians especially in cases in which different options including ICI are validated. Such data may also be useful for proposing clinical trials design of treatment intensification with a combination of an anti-PD(L1) with chemotherapy, targeted therapy or anti-CTLA4 for patients with high-volume disease and/or LM.

The aim of this study was to assess the prognostic impact of LM and MS number on response rate and long-term clinical outcomes (progression-free survival [PFS] and overall survival [OS]) in a large multicentric cohort of patients treated with single anti-PD(L)1 agent for different solid tumors. We also assessed the impact of ML type (liver vs. others MS) in patients with only one MS.

## 2. Materials and Methods

### 2.1. Patients

All patients with advanced or metastatic solid tumors who started a single-agent anti-PD(L)1 between January 2012 and October 2018 in one of the three sub-sites of the Lyon University hospital were included in the large retrospective IMMUCARE (Immunology Cancer Research) cohort of IC-HCL (Institut de cancérologie des Hospices Civils de Lyon), Lyon, France [14,15]. Patients with four different histologies (melanoma, non-small cell lung cancer [NSCLC], urothelial carcinoma, and clear cell renal carcinoma) were included. Single-agent Anti-PD(L)1 could be administered at any line of treatment according to guidelines for each different tumor subtype [16,17,18,19]. Only patients with single-agent anti-PD(L)1 were included in this study because there was a validated standard of care in France for the four types of cancer included in this study during the period of inclusion, and also to provide a homogeneity between all of the included patients. Indeed, between 2012 and 2018, nivolumab was approved in 2016 for first-line metastatic melanoma, in 2017 for second-line metastatic NSCLC and for second-line metastatic clear cell renal carcinoma.

Pembrolizumab was approved in 2016 for first-line metastatic melanoma, in 2017 for second-line metastatic NSCLC, in 2018 for metastatic urothelial carcinoma and for first-line metastatic NSCLC with a PDL1 expression of ≥50%. Atezolizumab was approved for second-line metastatic NSCLC in 2018. The exclusion criteria included patients treated with anti-CTLA4, patients treated with combination therapy (anti-PD(L)1 with anti-CTLA4, targeted therapy, or chemotherapy), patients treated for a localized cancer with a curative intent, patients included in a clinical trial and patients under the age of 18. This study has been approved by the ethical review board of the Hospices Civils de Lyon.

### 2.2. Data Collection

Data collection was performed retrospectively by three different investigators (PC, TR and MM) using a standardized data collection form. The data collected from electronic medical records included MS number at ICI initiation (1 or ≥2 and <3 or ≥3), MS type (liver vs. lung vs. brain vs. bone), age, gender, Eastern Cooperative Oncology Group (ECOG) performance status (PS), BMI, tumor histology (melanoma, NSCLC, or urologic cancer), >third line in metastatic setting, smoking habits, and plasmatic albumin level.

Patient follow-up was performed according to local clinical practice: clinical evaluation at each ICI administration and CT scan every 8 to 12 weeks. The anti-PD(L)-1 agents were administered intravenously according to the approved schedule for each molecule until unacceptable progression or toxicity occurred. An evaluation of the response to anti-PD(L)1 was based on the Response Evaluation Criteria in Solid Tumors (RECIST) 1.1 criteria and classified as a progressive disease (PD), stable disease (SD), partial response (RP) and complete response (CR). The date of disease progression was assessed by the treating physician according to the clinical symptoms or imaging evaluation.

### 2.3. Statistical Analyses

The characteristics of patients and clinical outcomes were separately analyzed according to the presence of LM (absent vs. present), MS number (1 or ≥2 and <3 or ≥3) and MS type (liver vs. lung vs. brain vs. bone). The cut-off points for MS number were based on those used in previous publications on this topic [20,21]. The Fisher exact test was used to compare binary or qualitative variables. The Mann–Whitney test was used to compare the quantitative variables. Response rate and radiologic progression were defined according to the RECIST 1.1 criteria.

The co-primary objective was to assess the impact of LM and MS number on progression-free survival (PFS) and overall survival (OS). The impact of the MS number was assessed using the thresholds of one and three MS. The secondary objective was to assess the response rate, PFS and OS for each MS type (liver vs. lung vs. brain vs. bone) in the subgroup of patients with only one MS. Progression-free survival was defined as the time from the start of anti-PD(L)1 agent to radiological or clinical progression or in-treatment death. Overall survival was defined as the time from the start of treatment to death from any cause. OS and PFS were assessed using the Kaplan–Meier method and compared between groups using two-tailed log-rank tests. We evaluated the impact of MS number, liver metastases and MS type on time-to-event endpoints. Adjusted hazard ratios were estimated using the multivariate Cox proportional hazards model, including the main confounding factors. All tests were two-sided and *p*-values below 0.05 were considered statistically significant. Statistical analyses were performed using R software.

## 3. Results

### 3.1. Patients Characteristics

In this cohort, a total of 759 patients treated with a single-agent anti-PD(L)1 were included (Figure 1). At the time of anti-PD(L)1 initiation, the main tumor type was NSCLC (*n* = 537, 71%) followed by melanoma (*n* = 144, 19%) and urologic cancer (urothelial carcinoma or clear cell renal carcinoma) (*n* = 78, 10%). Patients were mainly male (71%) with a median age of 66 years (19–94), 192 patients (26%) had PS ≥ 2, and 205 patients (27%) had received ≥3 lines of treatment in a metastatic setting. A total of 167 patients (22%) had LM, and a majority of them (86%) also had metastases in at least one other site. Overall, 370 patients (49%) had multiple MS and 389 patients (51%) had only one MS. Patients with at least two MS had a statistically significant PS ≥ 2 (31% vs. 23%, *p* ≤ 0.001) and more frequently had been previously treated with at least three lines of treatment (35% vs. 24%, *p* = 0.016) than patients with only one MS. Among patients with only one MS, 3% (*n* = 23) had only liver metastases, 6% (*n* = 49) only brain metastases, 16% (*n* = 124) only lung metastases and 8% (*n* = 62) only bone metastases. Among the 537 patients followed for lung cancer, 276 (51%) of them had multiple metastatic sites, and 261 (49%) had only one metastatic site. Table 1 summarizes patient characteristics according to number of metastatic sites (1 or ≥2).

### 3.2. Association between MS Number and Long-Term Outcomes

With a median follow-up was 16.6 months (95% CI, 15,3–17,6), patients with at least two MS had a shorter median PFS of 2.6 months (95% CI, 2.3–3.2 months) compared with 5.0 months (95% CI, 3.7–6.0 months) of patients with only one metastatic site (HR = 1.41; 95% CI: 1.19–1.67; *p* < 0.001). The median OS of patients with multiple MS was of 7.7 months (95% CI, 6.3–9.3 months) compared with 15 months (95% CI, 12.3–18.2 months) of patients with only one MS (HR = 1.63; 95% CI: 1.36–1.96; *p* < 0.001) (Figure 2). Results regarding OS were consistent in subgroups of patients with NSCLC and melanoma (Appendix A). The difference between median PFS according to metastatic sites (1 vs. ≥2) was statistically significant in the NSCLC cohort but not in the melanoma cohort.

Patients with at least three metastatic sites also had shorter PFS (HR = 1.39; 95% CI: 1.17–1.66; *p* = 0.00026) and OS (HR = 1.63; 95% CI: 1.35–1.96; *p* < 0.001) compared to the patients with less than three MS (Appendix A). Similar results were observed in subgroups of patients with NSCLC and melanoma (Appendix A). Furthermore, significant trends for both PFS and OS were observed between patients with different numbers of MS (1 MS versus 2 MS versus 3 MS versus >3 MS) in all cohort (Appendix A).

### 3.3. Association between Liver Metastases and Clinical Outcomes in All Patients

The presence of LM was associated with a shorter median PFS: 1.9 months (95% CI: 1.8–2.5) compared to 4.0 months (95% CI: 3.6–5.4) in patients without LM (HR = 1.63; 95% CI: 1.33–1.98; *p* < 0.001). The median OS of patients with LM was 5.2 months (95% CI: 4.0–7.7) compared with 12.8 months (95% CI: 11.2–15.1) of patients without LM (HR = 1.86; 95% CI: 1.51–2.28; *p* < 0.001) (Figure 3).

Analyses was adjusted with the presence of multiple MS, patient age, performance status, and tumor type, revealing that LM was an independent prognostic factor of PFS (adjusted HR = 1.57; 95% CI: 1.27–1.96; *p* < 0.001) and OS (adjusted HR = 1.77; 95% CI: 1.41–2.22; *p* < 0.001) (Table 2 and Appendix A).

### 3.4. Association between Liver Metastases and Clinical Outcomes in the Subgroup of Patients with Only One MS Involved

In the subgroup of patients with multiple MS, the presence of LM was also associated with shorter median PFS (HR = 1.67; 95% CI: 1.31–2.12; *p* < 0.001) and a shorter median OS (HR = 1.82; 95% CI: 1.41–2.34; *p* < 0.001). However, in the subgroup of patients with only one MS, median PFS (HR PFS = 0.95; 95% CI: 0.58–1.56; *p* = 0.85) and OS (HR OS = 0.89; 95% CI: 0.50–1.59; *p* = 0.68) were not statistically different according to the presence of LM (Figure 4). Indeed, the type of metastatic site (liver vs. brain vs. bone vs. lung) had no impact on PFS on OS in patients with only one MS. The overall response rate (ORR) was also not different according to MS type in patients with only one site of metastases (Figure 5).

## 4. Discussion

In this large retrospective study, we investigated the impact of the number of metastatic sites and liver metastases on ICI efficacy among patients with various solid tumors treated with single-agent anti-PD(L)-1. We showed that the presence of LM and multiple MS were associated with poor long-term survival outcomes. However, the prognostic impact of LM was not associated with shorter PFS or OS compared to other MS types (brain, bone or lung) in the subgroup of patients with only one MS. The results were consistent in the subgroups of patients treated for metastatic melanoma or NSCLC.

Before the advent of ICIs, the number and type of distant metastasis sites were investigated as prognostic factors in several cohorts of patients with different malignancies [8,10,22]. For NSCL, the majority of studies showed that liver or bone metastases are associated with poor outcomes. However, William Gibson et al. found that the number of metastatic sites were more important than MS type. This last result is consistent with Hendricks et al.’s study that showed that overall survival was significantly shorter in patients with more than three MS. For melanoma, the large retrospective study published by Balch et al. showed that the presence of at least three MS was also associated with poor outcomes and that lung metastases were associated with a significantly higher survival rate than metastatic melanoma in any other visceral site [23]. Similar results were observed in other malignancies, such as bladder cancer [24,25].

Several studies described the impact of MS number and type in cohorts of patients treated with ICIs and only two of them included patients treated with cancer of various histologies [12,13]. These two studies showed that LM were associated with shorter OS. In Botticelli et al.’s study, including 90 patients receiving ICIs in a phase I clinical trial, a MS number of ≥3 was also associated with shorter long-term survival outcomes [12]. These results support a recent study in NSCLC, showing that multiple MS and LM are associated with poorer response to ICIs, and that a combination strategy might effectively control LM [26]. However, a study by Pires da Silva et al. found that LM was the only significant factor associated with shorter overall survival in melanoma patients treated with the ipilimumab and nivolumab combination [27]. Our assessment of the impact of numbers of LM and MS is the largest cohort study ever published on this topic, including patients with several primary cancer types in a real-life setting. We confirmed that patients with LM and multiple sites of distant metastases experience inferior long-term clinical outcomes. Furthermore, this is the first study to show that patients with only one LM had a non-statistically different response rate and long-term clinical outcomes compared to other MS (lung, bone, or brain) in the subgroup of patients with only one MS. Consequently, liver metastases appear to be associated with poor outcomes only in patients with multiple metastatic sites.

The negative impact on prognosis of liver metastases could be partially explained by an immune tolerance induced by the immunosuppressive micro-environment of the liver. This concept has initially been described after the observation that allogenic liver transplantations were feasible using a non-histocompatible transplant [28]. Furthermore, the liver is a secondary lymphoid organ that contains a high density of regulatory T-cells as well as killer T-cells. A previous study pointed out that the liver contains partially activated CD8+ T-cells and has the ability to trap activated CD8+ T-cells [29,30]. Liver metastases of melanoma and NSCLC show lesser marginal CD8+ T-cell infiltration compared to other metastatic sites, suggesting a possible lower efficacy of ICIs in the presence of liver metastases [31]. Finally, Qiao et al. showed that liver metastases have lower PDL1 expression on CD8+ T-cells compared to other metastatic sites [26]. Interestingly, the immune tolerance of the liver was indirectly assessed by Facciorusso et al., who described the negative impact of a low lymphocyte-to-monocyte ratio (LMR) after radiofrequency ablation for colorectal liver metastases [32].

In a metastatic liver microenvironment, resident cells such as hepatocytes, sinusoidal cells, and Kupffer cells exhibit tumor progression-promoting activities and mediated immunosuppression. This mechanism-involved production of angiogenesis factors, such as VEGF, which promotes tumor progression and decreases the infiltration of CD8+ T-cells in liver metastases [33]. These observations partially explained the poor outcomes associated with liver metastases as described in our study, but did not explain why the presence of metastases only in the liver was not associated with poorer response to ICIs. It could be interesting to compare the biomarkers associated with ICI responses, such as CD8+ T-cells infiltration and PDL1 expression on CD8+ T-cells in liver metastases between patients with only liver metastases and patients with multiple metastatic sites.

Our study has some limitations. First, despite a large number of included patients, it remains a retrospective cohort. In addition, some data are missing—for example, the presence and the prognostic impact of lymph node metastases. It would be interesting to consider this factor particularly in the melanoma subgroup because it is the first metastatic relay in this pathology and studies show that lymph node invasion alone is associated with a better prognosis than visceral metastases [34]. There is also a lack of information on the prior local treatment of brain metastases (neurosurgery or radiotherapy) because our study isolated brain metastases were not associated with poorer prognosis, likely because they were largely controlled using a local treatment. Moreover, some subgroups and especially the subgroup of patients with LM alone (3%, *n* = 24) are relatively small in size compared to the overall population. Consequently, additional studies are needed to confirm that patients with LM alone have similar long-term outcomes than patients with other MS (lung, brain, and bone) among patients with a unique MS. Although we investigated for the first time the impact of MS number and LM in four different malignancies (NSCLC, melanoma, urothelial carcinoma, and renal carcinoma), anti-PD(L)1 are currently used in many other kinds of cancer treatment, such as head and neck tumors, colorectal cancers, or hepatocellular carcinoma [35,36]. Future studies are needed to confirm our results in cohorts including more cancer subtypes. Finally, we only included patients treated with a single-agent anti-PD(L)1, but anti-CTLA4 drugs such as ipilimumab are now used alone or in combination with anti-PD(L)1 in metastatic melanoma, NSCLC, or renal clear cell carcinoma, and it could be interesting to evaluate the impact of MS number and LM in patients treated with anti-CTLA4, especially in combination [37]. Furthermore, it was not possible to assess if the outcomes were different regarding the type of ICIs used (anti-PDL1 versus anti-PD1) due to the low number of patients treated with anti-PDL1 (n = 8) compared to patients treated with anti-PD1. However, previous studies showed that efficacy of the ICIs are different between anti-PD1 and anti-PD(L)1, and it could be interesting to assess if efficacy differences on liver metastases exist between each type of ICI.

While there is a clear negative effect of LM on long-term survival outcomes, this trend is probably not unique to the liver as suggested by the fact that patients with LM alone had similar outcomes of patients with only one other involved MS. Indeed, the presence of other tumor characteristics not evaluated in this study such as PDL1 expression, TMB or presence of TILs also have an impact on the anti-PD(L)1 response. In future studies, it could be interesting to compare these tumor characteristics between LM and the other MS.

The use of anti-PD(L)1 has been associated with improved outcomes in previously published randomized controlled trials and is now the standard of care for all the cancer types included in this study [16,17,18,19]. However, as only patients receiving immunotherapy were included in this study, we can only describe the factors associated with oncologic outcomes while receiving immunotherapy. It is true that we could not conclude whether our findings are specific or not to immunotherapy, and if similar findings would have been obtained with other treatments, such as chemotherapy or targeted therapy.

Our study suggests that ICIs should be introduced as early as possible before the development of metastases in different organs. This is especially true given the fact that ICIs frequently lead to immune-related adverse events and should be used with patients with the highest probability of response [38]. The utilization of a combination therapy of ICIs with targeted therapy or chemotherapy could be an option to have a maximum effect in patients with liver metastases. Notably, the results of the KEYNOTE-189 study showed that pembrolizumab associated with chemotherapy prolonged OS in patients with liver metastases [39]. Furthermore, in the IMpower-150 study, NSCLC patients with liver metastases had poor outcomes with a combination of ICI and chemotherapy but seemed to have better outcomes when anti-angiogenic drugs were added to treatment [40]. Consequently, future studies are needed to identify the best treatment strategies for this subgroup of patients.

## 5. Conclusions

The identification of biomarkers associated with responses to anti-PD(L)-1 remains a primary objective. Our study showed that the presence of multiple sites of distant metastases and liver metastases are associated with poor outcomes in a large cohort of patients treated with ICIs for different malignancies. The mechanisms explaining the poor prognoses associated with liver metastases should be studied further.

## Figures and Tables

**Figure 1 biomedicines-11-00083-f001:**
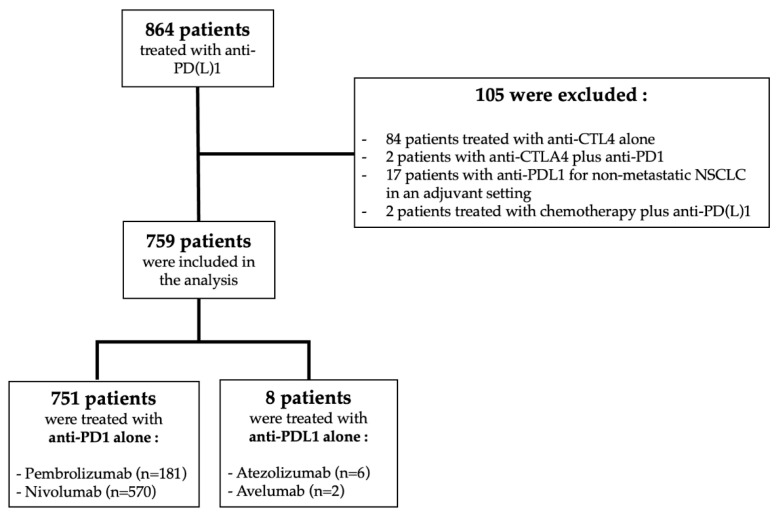
Study flow chart.

**Figure 2 biomedicines-11-00083-f002:**
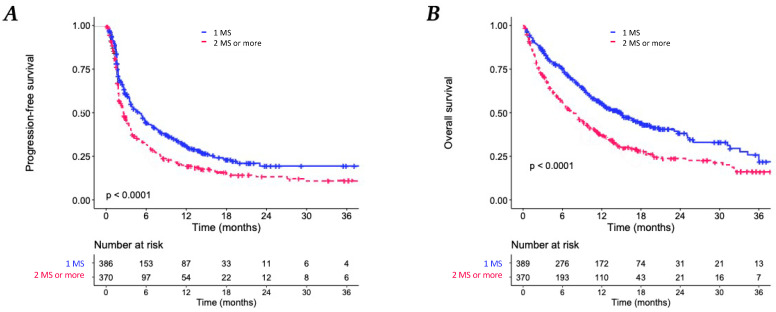
Kaplan–Meier analyses. (**A**) Progression-free survival (PFS) according to metastatic site (MS) number (1 MS versus ≥2 ML). (**B**) Overall survival according to MS number (1 MS versus ≥2 MS).

**Figure 3 biomedicines-11-00083-f003:**
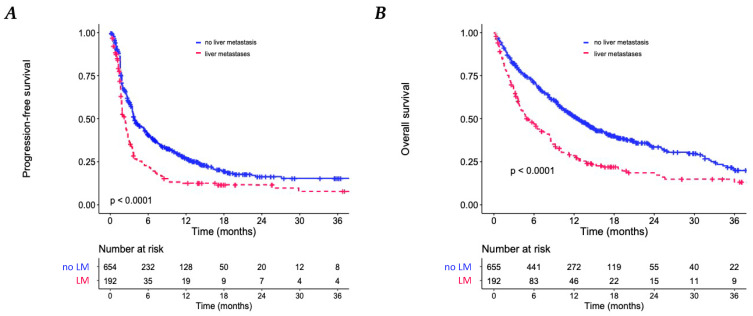
Kaplan–Meier analyses. (**A**) Progression-free survival (PFS) according to the presence of liver metastases (LM). (**B**) Overall survival according to the presence of LM.

**Figure 4 biomedicines-11-00083-f004:**
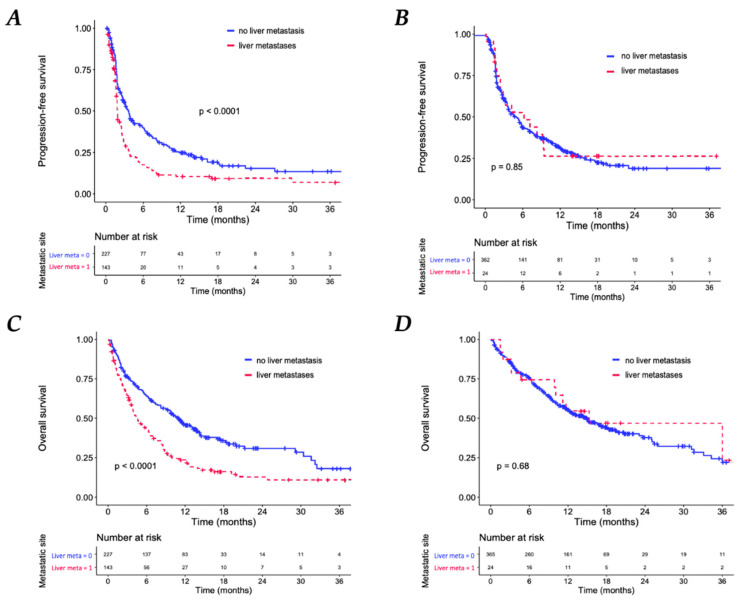
Kaplan–Meier Analyses. Progression-free survival (PFS) according to the presence of liver metastases in patients with multiple sites of metastasis, (**A**) and in patients with only one MS (**B**). Overall survival (OS) according to the presence of liver metastases in patients with multiple sites of metastasis (**C**) and in patients with only one MS (**D**).

**Figure 5 biomedicines-11-00083-f005:**
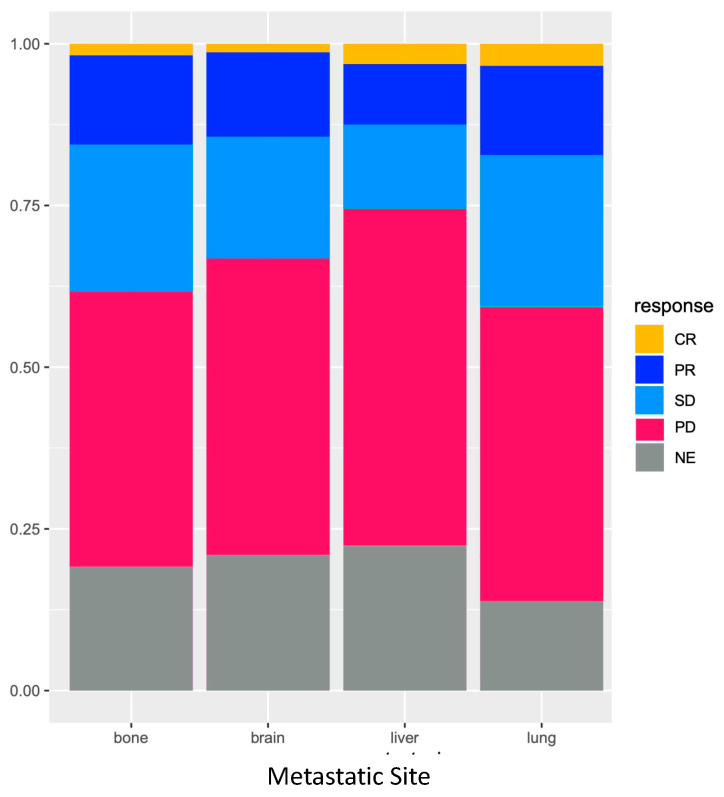
Overall response rate (ORR) according to metastatic location type in the subgroup of patients with only one site of metastasis. Legend: CR—Complete response; PR—Partial response; SD—Stable disease; PD—Progressive disease; NE—Non-evaluable.

**Table 1 biomedicines-11-00083-t001:** Patient characteristics according to the number of metastatic sites (one MS versus two MS).

Variable	All (n = 759)			
	All Patients	Patients with 1 MS (n = 389)	Patients with ≥2 MS (n = 370)	*p*
Age, years, median (25th–75th) NA = 0	66 (58–73)	67 (60–74)	66 (58–72)	0.011
Gender male (%) NA = 0	536 (71%)	276 (71%)	260 (70%)	0.90
PS ≥ 2 (%) NA = 33	192 (26%)	77 (20%)	115 (33%)	0.00015
BMI (%) NA = 7 <18 18–30 >30	60 (8%) 597 (79%) 95 (13%)	32 (8%) 297 (77%) 57 (15%)	28 (8%) 300 (82%) 38 (10%)	0.17
Primary tumor (%) NA = 0 NSLC Melanoma Urologic	537 (71%) 144 (19%) 78 (10%)	261 (67%) 73 (19%) 55 (14%)	276 (75%) 71 (19%) 23 (6%)	0.0014
≥3rd line in metastatic setting (%) NA = 0	205 (27%)	90 (23%)	115 (31%)	0.016
Type of metastasis (%) NA = 0 Brain only Lung only Liver only Bone only Other only Multiple sites	49 (6%) 124 (16%) 23 (3%) 62 (8%) 131 (17%) 370 (49%)	_	_	-
Any history of autoimmune disorder (%) NA = 12	71 (10%)	44 (12%)	27 (7%)	0.091

**Table 2 biomedicines-11-00083-t002:** Overall survival prognostic factors in univariate and multivariate analysis (Cox model).

Characteristics	N (%)	Overall Survival				
		Median OS (95% CI)	Unadjusted Analysis		Adjusted Analysis	
			HR (95% CI)	*p*	HR (95% CI)	*p*
Age NA = 0 <70 ≥70	489 (64%) 270 (36%)	10.3 (9.4–12.8) 11.1 (8.5–13.8)	REF 1.0 (0.84–1.23)	0.85	REF 1.14 (0.94–1.39)	0.18
PS < 2 (%) NA = 33 PS ≥ 2	534 (74%) 192 (26%)	14.4 (12.8–16.7) 3.6(3.1–5.0)	0.36 (0.29–0.43) REF	<0.0001	0.40 (0.33–0.49) REF	<0.0001
Primary tumor (%) NA = 0 NSLC Melanoma Urologic	537 (71%) 144 (19%) 78 (10%)	9.3 (8.3–10.9) 25.4 (16.2-NA) 10.5 (7.3-NA)	REF 0.47 (0.36–0.63) 0.84(0.61–1.15)	<0.0001	REF 0.52 (0.39–0.70) 0.95 (0.68–1.32)	<0.0001
Nb of metastatic site 1 >1	389 (51%) 370 (49%)	15.0 (12.3–18.2) 7.7 (6.3–9.3)	REF 1.63 (1.36–1.96)	<0.0001	REF 1.28 (1.04–1.57)	0.021
Liver metastasis No Yes	592 (78%) 167 (22%)	17 (14-NA) 19 (12-NA)	REF 1.86 (1.51–2.28)	<0.0001	REF 1.77 (1.41–2.22)	<0.0001

## Data Availability

The data presented in this study are available on request from the corresponding author. The data are not publicly available due to privacy.

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
