# Peer review of "Impact of Liver Metastases and Number of Metastatic Sites on Immune-Checkpoint Inhibitors Efficacy in Patients with Different Solid Tumors: A Retrospective Study"

_biomedicines, 2022, doi:10.3390/biomedicines11010083_

Round 1
Reviewer 1 Report
See the attached document.

Author Response
Our responses are in the attached document.
Comments and Suggestions for Authors : The article titled "Impact of Liver Metastases and Metastatic Sites Number on Immune-Checkpoint Inhibitors Efficacy in Patients with Different Solid Tumors" shows a retrospective study on predictive biomarkers of response to immunotherapy in patients with solid tumors. The two predictive factors analyzed are the presence of liver metastases and the metastatic sites number. The study is correctly written and structured, being its simple reading and the reader can get a quick idea of the conclusions that the authors have reached. However, the article presents a series of deficiencies that should be corrected to consider its publication in the journal. The statistical analysis carried out by the authors contains a series of confounding factors that do not allow to validate the conclusions, and therefore, although the approach of the article is correct, the method to reach the conclusions is not the correct one from my point of view. The main problem that is found when reading the article is that the two predictive factors analyzed are also prognostic factors, and this makes the statistical analysis must be more complex than that carried out by the authors. Therefore, the authors should carry out a restructuring of the article, with a new statistical analysis that allows to eliminate these confounding factors.In addition to the above, the use of the language is correct and does not need corrections or revision. Likewise, the references used are mostly current and make a good reflection of the literature on the subject being addressed. The figures and tables in the article (except for one of them) do not need corrections and allow a quick reading of the general characteristics of the patients included in the study and the results obtained. The most interesting and valuable section of the article is the discussion. At this point the authors make a great comparison with the current literature, making a very correct analysis of the results that have been found, as well as the limitations which increases the value of the article.
The following changes are what I would suggest for the article:
Major changes
- Results: as previously indicated, the authors should carry out a new statistical analysis so that the conclusions they have reached can be valid. A logistic regression could be performed where all the confounding factors that the article might have could be analyzed. The mere presence of liver metastases and having multiple locations of metastatic ones is associated with a worse prognosis in all types of tumors and therefore immunotherapy in these patients will have worse results. Having liver lesions is an unfavorable prognostic factor already known and described in the literature, therefore, statistical analysis should consider this fact.
We totally agree with the reviewer that confounding factors exist and must be taken into account in such retrospective study. That’s why we have built a multivariate model (Cox proportional hazards models) including the main confounding factors : ECO-PS, age, number of metastatic site, type of cancer and presence of liver metastases. The results of our multivariate analysis showed that both liver metastases and metastatic site number are independent prognostic factors on overall survival. The results of this analysis were already included in the manuscript on Table 2.
Table 2. Overall survival prognostic factors in univariate and multivariate analysis (Cox model)
|
Characteristics |
N (%) |
Overall survival |
||||
|
median OS (95% CI) |
Unadjusted analysis |
Adjusted Analysis |
||||
|
HR (95%CI) |
P |
HR (95%CI) |
P |
|||
|
Age NA=0 < 70 ≥70 |
489 (64%) 270 (36%) |
10.3 (9.4-12.8) 11.1 (8.5-13.8) |
REF 1.0 (0.84-1.23) |
0.85 |
REF 1.14 (0.94-1.39) |
0.18 |
|
PS < 2 (%) NA=33 PS ≥ 2 |
534 (74%) 192 (26%) |
14.4 (12.8-16.7) 3.6(3.1-5.0) |
0.36 (0.29-0.43) REF
|
<0.0001 |
0.40 (0.33-0.49) REF
|
<0.0001 |
|
Primary tumor (%) NA=0 NSLC Melanoma Urologic |
537 (71%) 144 (19%) 78 (10%) |
9.3 (8.3-10.9) 25.4 (16.2-NA) 10.5 (7.3-NA) |
REF 0.47 (0.36-0.63) 0.84(0.61-1.15) |
<0.0001 |
REF 0.52 (0.39-0.70) 0.95 (0.68-1.32) |
<0.0001 |
|
Nb of metastatic site 1 >1 |
389 (51%) 370 (49%) |
15.0 (12.3-18.2) 7.7 (6.3-9.3) |
REF 1.63 (1.36 – 1.96) |
<0.0001 |
REF 1.28 (1.04-1.57) |
0.021 |
|
Liver metastasis No Yes |
592 (78%) 167 (22%) |
17 (14-NA) 19 (12-NA) |
REF 1.86 (1.51 - 2.28) |
<0.0001 |
REF 1.77 (1.41 - 2.22) |
<0.0001 |
To answer the important question of the predictive impact of liver metastases (LM) or metastatic site (MS) number on checkpoint inhibitors efficacy, a randomized controlled trial or a meta-analysis of such randomized trials should be conducted in order to compared the efficacy of checkpoint inhibitors to a chemotherapy across stratas. As our study does not include a control arm, it was not possible to perform an interaction test to assess the predictive impact of liver metastases or metastatic site numbers on anti-PD(L)1 efficacy relative to other therapies.
For all these reasons, we can only conclude in our study that the presence of liver metastases and metastatic site numbers have a prognostic impact oamong patients treated with anti-PD(L)1.
- Abstract: it must be modified integer because it is a very uninformative abstract. In the article the section of material and methods is of great value, however, in the abstract it does not allow a quick reading on how the article has been made.
We want to thanks the reviewer for this important comment. We modified the section methods of the abstract as follow :
Background : ICIs have dramatically improved patient outcomes in different malignancies. However, the impact of liver metastases (LM) and metastatic sites (MS) number remains unclear in patients treated with single agent anti-PD(L)1.
Methods : We aimed to assess the prognostic impact of LM and MS number on progression-free survival (PFS) and overall survival (OS) in a large single arm retrospective multicentric cohort (IMMUCARE) of patients treated with anti-PD(L)-1 for different solid tumors.
Results A total of 759 patients were enrolled from January 2012 to October 2018. The primary tumor types were non-small cell lung cancer (71%), melanoma (19%), or urologic cancer (10%). At time of ICI initiation, 167 patients (22%) had LM and 370 patients (49%) had more than 1 MS. LM was associated with a shorter median PFS of 1.9 months (95%CI : 1.8-2.5) vs 4.0 months (95%CI : 3.6-5.4) in patients without LM (P<0.001). The median OS of patients with LM was of 5.2 months (95%CI : 4.0-7.7) compared with 12.8 months (95%CI : 11.2-15.1) (P<0.001). Interestingly, LM were not associated with shorter PFS or OS compared to others MS type (brain, bone or lung) in patients with only one MS. Patients with multiple MS had also poor clinical outcomes compared to patients with only one MS. Presence of LM and MS number were independent prognostic factor on overall survival.
Conclusion : Presence of LM or multiple MS were associated with poorer survival outcomes in patients treated with anti-PD(L)-1.
Minor changes
- Title: the authors must indicate the type of study that has been carried out, in this acase a retrospective study.
We want to thank the reviewer for this comment. We indicated in the title that it’s a retrospective study :
Impact of Liver Metastases and Metastatic Sites Number on Immune-Checkpoint Inhibitors Efficacy in Patients with Different Solid Tumors : a retrospective study
- Keywords: add "predictive biomarkers".
Thank you for this important comment. We added “prognostic biomarkers” as a keyword rather than “predictive biomarkers” as we explained in our response to your first comment :
Keywords: Immune checkpoint inhibitors; PD1 inhibitors; PDL1 inhibitors; Metastatic sites; Liver metastases, prognostic biomarkers.
- Introduction: a paragraph should be added explaining the mechanism of action of immunotherapy and immune checkpoint inhibitors.
We added in the introduction section this following sentence to explain the mechanism of action of immune checkpoint inhibitors :
PD-1 is a key regulator of the threshold of immune response and peripheral immune tolerance. It is expressed on activated T cells, B cells, macrophages, regulatory T cells (Treg) and natural killers (NK) cells. Binding of PD-1 to its ligands (PDL1 or PDL2) which are frequently expressed on tumor cells, results in suppression of proliferation and immune response of T cells. Consequently, activation of PD-1/PD-L1 signaling serves as a principal mechanism by which tumors evade antigen-specific T-cell immunologic responses [1].
- Introduction, line 61-63: add the ECOG as a response factor. In the past, PD-L1 the other well-known predictive factor of response to immunotherapy is the patient's performance status.
We are totally agree with the reviewer and we added in the introduction section of our manuscript the following sentence :
As a consequence, some clinical features such as ECOG performance status are used in routine practice to predict response to ICIs.
- Introduction, line 75: delete that it is a retrospective study. It is already clearly indicated in material and methods.
As suggested we deleted that is a retrospective study at the end of the introduction section :
The aim of this retrospective study was to assess the impact of LM and MS number on response rate and long-term clinical outcomes (progression-free survival [PFS] and overall survival [OS]) in a large multicentric cohort of patients treated with single anti-PD(L)1 agent for different solid tumors.
- Results, line 144: it was indicated in material and methods that renal patients were
also collected. Were they not subsequently analyzed? Please specify.
For simplicity and because of the low number of urothelial carcinoma and clear cell renal carcinoma included in our study we decided to integrated this two types of cancer under the term of “urologic cancer”. We specified that point in the result section of the manuscript as suggested by the reviewer :
In this cohort, a total of 759 patients treated with anti-PD(L)1 single agent were included (Figure 1). At the time of anti-PD(L)1 initiation, the main tumor type was NSCLC (n = 537, 71%) followed by melanoma (n= 144, 19%) and urologic cancer (urothelial carcinoma or clear cell renal carcinoma) (n = 78, 10%).
- Figure 1: In the figure you can see an underline in red that should not be provided.
We removed all the underlines in red from the Figure 1.
- Table 1: If collected for the study, indicate the number of average doses that were
applied to patients.
Thank you for this important comment. Unfortunately we did not collect this data and it is consequently not possible to add it in the Table 1
Reviewer 2 Report
Very interesting paper. I have only some (minor) suggestions.
1) How were the KM curves compared? I assume through then log-rank test but this should be specified in the statistical chapter.
2) I suggest to add some comments in the Discussion on the inflammatory prognostic biomarkers in patients with liver metastases (cite the paper PMID: 27122671)
Author Response
Our response to reviewers are in the attached document
Very interesting paper. I have only some (minor) suggestions.
1) How were the KM curves compared? I assume through then log-rank test but this should be specified in the statistical chapter.
We want to thank the reviewer for this very important comment regarding the statistical chapter that we modified as follow as suggested :
OS and PFS were assessed using Kaplan-Meier method and compared between groups using 2-tailed log-rank tests.
2) I suggest to add some comments in the Discussion on the inflammatory prognostic biomarkers in patients with liver metastases (cite the paper PMID: 27122671)
As suggested by the reviewer we added the following sentence in the discussion section of our manuscript :
Interestingly, the immune tolerance of the liver was indirectly assessed by Facciorusso et al. that have described the negative impact of a low lymphocyte-to-monocyte ratio (LMR) after radiofrequency ablation for colorectal liver metastases [32]
We also added the corresponding refence :
- Facciorusso, A.; Del Prete, V.; Crucinio, N.; Serviddio, G.; Vendemiale, G.; Muscatiello, N. Lymphocyte-to-monocyte ratio predicts survival after radiofrequency ablation for colorectal liver metastases. World J Gastroenterol 2016, 22(16):4211-8, doi: 10.3748/wjg.v22.i16.4211.
